# HTS and PCR Methods Are the Most Used in the Diagnosis of Aspergillosis: Advantages over Other Molecular Methods

**DOI:** 10.3390/jof11100720

**Published:** 2025-10-06

**Authors:** Carlos Alberto Castro-Fuentes, Esperanza Duarte-Escalante, María Guadalupe Frías-De-León, María del Carmen Auxilio González-Villaseñor, María del Rocío Reyes-Montes

**Affiliations:** 1Posgrado en Ciencias Biológicas, Facultad de Medicina, Universidad Nacional Autónoma de Mexico, Mexico City 04510, Mexico; castrofuenca@gmail.com; 2Departamento de Microbiología y Parasitología, Facultad de Medicina, Universidad Nacional Autónoma de Mexico, Avenida Universidad 3000, Ciudad Universitaria, Coyoacán, Mexico City 04510, Mexico; dupe@unam.mx; 3Unidad de Investigación, Hospital Regional de Alta Especialidad de Ixtapaluca, Servicios de Salud del Instituto Mexicano de Seguro Social para el Bienestar (IMSS-BIENESTAR), Carretera Federal Mexico-Puebla Km 34.5, Ixtapaluca 56530, Mexico; magpefrias@gmail.com; 4Instituto de Biología, Universidad Nacional Autónoma de Mexico, Mexico City 04510, Mexico; mcgv@ib.unam.mx

**Keywords:** aspergillosis, diagnosis, molecular method, NGS, NTG, PCR, ddPCR, PCR-RFLP, LAMP, U-dHRM

## Abstract

Aspergillosis includes a variety of diseases caused by species of the genus *Aspergillus*, ranging from non-invasive allergic diseases to chronic, invasive pulmonary infections, which are potentially fatal in immunocompromised hosts. Therefore, there is an urgent need for new diagnostic tools and the optimization of existing tests to improve patient care. This work reviews the most commonly used molecular methods for the diagnosis of aspergillosis from clinical samples, emphasizing their advantages. These methods included HTS, NTS, ISH, microarrays, PCR-RFLP, LAMP, and PCR in various modalities (qPCR, multiplex PCR, nested PCR, RT-PCR, endpoint PCR, U-dHRM, and ddPCR). The review showed that the most commonly used methods for diagnosing aspergillosis are NGS and PCR in their different modalities; however, each method has advantages and disadvantages. qPCR is the method that has demonstrated the greatest sensitivity and specificity on clinical samples (such as blood, serum, bronchoalveolar lavage [BAL], tissue, or sputum), since it detects specific sequences, and the validation of this method shows greater progress in achieving this objective. Likewise, NGS showed that BAL is the most suitable sample, with a higher fungal load than sputum or blood. On the other hand, NGS is not a targeted technique, since it sequences all the genetic material present. Additionally, the sensitivity for detecting pathogens decreases when clinical samples are used due to the high background of nucleic acids present in the human host.

## 1. Introduction

The genus *Aspergillus* is widely distributed in the environment, including in outdoor and indoor spaces and hospital settings, particularly in the latter. Its presence has been associated with building renovation and construction [1]. Additionally, it is considered one of the leading agents that cause opportunistic fungal infections. The genus *Aspergillus* is subdivided into 6 subgenera, 27 sections, and 75 series, with 453 accepted species [2,3]. Of these sections, at least 12 sections (*Fumigati*, *Flavi*, *Nigri*, *Terrei*, *Nidulantes*, *Usti*, *Aspergillus*, *Circumdati*, *Flavipides*, *Tanneri*, *Candidi*, and *Restricti*) contain clinically relevant species [4,5,6].

Likewise, aspergillosis encompasses a wide spectrum of clinical manifestations, including the colonization of pre-existing cavities, allergic and cutaneous forms, otomycosis, and invasive aspergillosis (IA) [7]. The latter is the most severe clinical form caused by *Aspergillus* species in susceptible hosts with compromised immune systems, with a mortality rate ranging from 50% to 95% [2,4,5,6,7,8,9,10]. It is essential to note that the species most frequently associated with IA are *A. fumigatus*, *A. flavus*, *A. niger*, and *A. terreus*; however, in recent years, several cryptic species have been identified in complicated cases of IA [9]. The increase in the number of immunosuppressed patients in recent years, associated with AI, requires a rapid and reliable diagnosis that includes the correct identification of the pathogen; however, this represents a challenge due to the number of cryptic species in the different sections [11]. Therefore, there is a need to identify the fungus at the species level, particularly because the new antifungal agents available differ in their spectrum of activity [12].

Molecular biology methods have also been used in the typing of *Aspergillus* isolates to identify and characterize pathogenic microorganisms that cause infectious outbreaks, the source of infection, and the pattern of dissemination. Among these methods are random amplified polymorphic DNA (RAPD), restriction fragment length polymorphism (RFLP), single-strand confirmation polymorphism (SSCP), amplified fragment length polymorphism (AFLP), multilocus microsatellite typing (MLMT), and multi-locus sequence typing (MLST) [13,14,15], in addition to high-throughput sequencing (HTS) [next-generation sequencing (NGS) and third-generation sequencing (TGS)] [16,17,18]. Molecular assays have also been used for the direct detection of *Aspergillus* DNA in clinical samples at the genus or species level, such as RFLP. In this method, an amplicon obtained through PCR is treated with a restriction enzyme that cuts the DNA at a specific restriction site, known as a recognition site, to generate several DNA fragments of varying sizes. These fragments are then subjected to electrophoresis, which allows bands of different sizes to be visualized [19]. Microarrays involve the hybridization of short probes that bind to sequences in specific regions of the sample’s DNA, placed on a solid support. These probes, upon binding, emit fluorescence, which is detected by an analyzer. DNA microarray detection allows for the simultaneous analysis of a large number of nucleic acid sequences from multiple pathogens in a single assay. The loop-mediated isothermal amplification (LAMP) method [20] is based on autocycling and high levels of DNA strand displacement activity mediated by the Bst polymerase of *Geobacillus stearothermophilus*, under isothermal conditions (60–65 °C) [21]. This assay involves two types of amplification reactions: template amplification from a loop structure formed at the 3′-terminal end and the ligation and elongation of new primers from the loop region. The final results of LAMP can be detected by turbidimetry, fluorometry, and colorimetry, making it more suitable for in situ testing [22]. Furthermore, in situ hybridization (ISH) is a method that detects the presence of a nucleotide sequence in fixed cells or tissues using a fluorochrome-labeled probe, which is complementary to the sequence to be identified and emits fluorescence that can be observed under a microscope. The complementarity between nucleotide sequences underlies the specificity of this technique. PCR in different modalities: Quantitative PCR (qPCR) differs from conventional PCR in that it monitors the amplification of the target DNA molecule during each amplification cycle, rather than at the end. DNA markers or fluorescent probes are added to the same reaction tube; in this way, the appearance of specific components is measured during amplification, making it possible to collect and analyze data simultaneously. Multiplex PCR involves using several pairs of primers, each specific to a particular *Aspergillus* species, in a single PCR reaction. The multiplex qPCR method has advantages over conventional qPCR, allowing for the simultaneous detection of several fungi in a single reaction and reducing the use of PCR samples [23]. The application of multiplex real-time PCR requires some technical considerations. First, the primer sets must be highly specific to the target species for accurate detection. Second, each primer set must have approximately the same melting temperature (Tm) because any differences in their Tm can cause bias in DNA amplification. Third, non-competitive primers and probes must be designed and used for the reactions. When more species are selected in a single reaction, the chances of generating primer dimers and the fluorescent probe competition increase. Finally, the lengths of the target sequences must be similar between target species because the length of the DNA fragment influences the amplification efficiency and the decay rate of environmental DNA [24]. Another type of PCR is nested PCR. In this technique, although PCR is very sensitive, when the amount of DNA is very small, a second pair of primers is added in the first amplification to carry out a second amplification, in which the product amplified in the first PCR is used as the target. A modification of this technique is “semi-nested PCR” (snPCR), in which, in the second PCR amplification, one of the primers from the first amplification is kept and a new one is added, which shortens the amplified region and detects more specific—generally species-specific—fragments. Reverse transcriptase-PCR (RT-PCR) allows for the use of RNA as a template to generate complementary DNA (cDNA). Using the reverse transcriptase enzyme, a single-stranded copy of cDNA is generated. This can then be amplified by a DNA polymerase, generating double-stranded cDNA that is fed into a standard PCR-based amplification process [25]. Endpoint PCR starts from a DNA template; then, a polymerase enzyme incorporates nucleotides complementary to this template DNA, resulting from the binding of primers to the template. The process consists of three phases: denaturation, hybridization, and elongation, which are performed at different temperatures. The three phases constitute one PCR cycle. Another recently described and broad-spectrum technique is the universal digital high-resolution melt (U-dHRM), which enables the detection of multiple pathogens at the single-genome level through the amplification of universal adapters to amplify all the sequences present, without the need for prior knowledge of them. This is followed by the high-resolution melting of the obtained amplicons and the implementation of a melting analysis in digital format that allows for identification and counting at the single-genome level, even in polymicrobial samples, and that eliminates competition for template amplification and efficiency biases. This format generates extensive melting curve training data, harnessing the potential of machine learning through big data for automated melting curve identification and allowing for the rapid identification and quantification of the sequences of all common pathogens in each sample [26]. The droplet digital PCR (ddPCR) technique enables the detection of specific DNA from infectious microorganisms with high sensitivity, high precision, and absolute quantification. The basis of this technique is the partitioning of the PCR reaction mixture into segregated droplets that are a thousand times smaller, allowing the amplification of the target of interest within each droplet, which is subsequently quantified by fluorescence. Based on the emitted fluorescence signal, amplification indicates a positive result, whereas the absence of fluorescence yields a negative result [27].

Other molecular methods that can be used include NGS and TGS. NGS consists of platforms that produce a large number of short reads (25–400 bp) of DNA sequences [28]. The most important platforms are Roche 454, Ion Torrent, and Illumina. The library preparation is carried out in an analogous manner, but the 454, SOLiD, and Ion Torrent platforms use emulsion PCR, while the Illumina platform uses solid-phase PCR [29]. These platforms can be used to obtain sequences of complete genomes or sequences restricted to genetic regions of interest; among these, the company Illumina Inc. stands out [30,31,32,33,34]. This method is an emerging technology that involves the parallel sequencing of several short DNA fragments. It is achieved through the following steps: (1) segmenting the DNA into several fragments; (2) labeling the DNA using primers or adapters that indicate the starting point for replication; (3) amplifying the adapter-tagged DNA fragments using PCR-based methods; (4) sequencing or reading the DNA fragments—in this step, different sequencing platforms (Illumina and pyrosequencing) can be used; (5) performing a computational alignment with a reference genome for the rapid detection, characterization, and genotyping of pathogens in clinical diseases of an unknown etiology [35]; and (6) reconstructing the complete sequence using reference sequences and exporting to data storage files [36,37].

Likewise, TGS technologies allow for read lengths of up to 2.3 Mb and do not require preliminary amplification, reducing the incidence of epigenetic mark loss [38]. There are two widely used vendors for long-read sequencing: Pacific Biosciences (PacBio) and Oxford Nanopore Technologies Inc. (ONT). PacBio can generate HiFi (high-fidelity) reads with a high base resolution. Oxford Nanopore can generate reads up to 2 Mb [39]. Additionally, other emerging TGS platforms, such as Molecu and 10X Genomics, rely on the assembly of short reads to generate synthetic long reads [40]. Oxford Nanopore’s technology uses nanopore sequencing, which uses the electrical profile of each DNA and RNA nucleotide for identification [41]. DNA molecules pass through a small pore, and the change in electrical current as the DNA passes through the pore allows DNA bases to be identified in real time [38]. The basis of ONT sequencing allows for the direct reading of RNA sequences; it is the only technology with this capability [42,43].

Furthermore, given the diversity of approaches offered by molecular techniques to identify microorganisms, the objective of this work was to review the most commonly used molecular methods for the diagnosis of aspergillosis from clinical samples (Figure 1).

## 2. Materials and Methods

A review was conducted on the molecular diagnosis of aspergillosis, with an emphasis on the clinical samples used in each of the molecular methods. The search was conducted in the Scopus, PubMed, ScienceDirect, MEDLINE, and SciELO databases using the following keywords: aspergillosis, aspergillosis diagnosis, molecular diagnosis in aspergillosis, PCR aspergillosis, NGS aspergillosis, NTS aspergillosis, microarrays aspergillosis, ISH aspergillosis, and RFLP aspergillosis.

The information obtained from the reviewed articles was organized in tables with the molecular methods in chronological order, indicating the species identified, the clinical form, the clinical sample, the sensitivity and specificity of the method, and the target used in the case of PCR.

## 3. Results

A total of 102 articles were included, of which 30 articles were related to the diagnosis of aspergillosis using the NGS method, 2 were related to the NTS method, 8 were related to the ISH method, 3 were related to the microarray method, 1 was related to the RFLP method, 4 were related to LAMP, 28 were related to qPCR, 12 were related to multiplex PCR, 11 were related to nested PCR, 1 was related to RT-PCR, 2 were related to endpoint PCR, 1 was related to U-dHRM, and 1 was related to ddPCR (Appendix A [44,45,46,47,48,49,50,51,52,53,54,55,56,57,58,59,60,61,62,63,64,65,66,67,68,69,70,71,72,73,74,75,76,77,78,79,80,81,82,83,84,85,86,87,88,89,90,91] and Appendix A [26,27,92,93,94,95,96,97,98,99,100,101,102,103,104,105,106,107,108,109,110,111,112,113,114,115,116,117,118,119,120,121,122,123,124,125,126,127,128,129,130,131,132,133,134,135,136,137,138,139,140,141,142,143,144,145].

Appendix A presents the molecular methods used for diagnosing aspergillosis, including NGS, NTS, ISH, microarrays, RFLP, and LAMP, along with information on the identified species, clinical presentation, biological samples used, sensitivity, specificity, and detection rate values, as well as references. Appendix A shows information on articles related to PCR methods used for the diagnosis of aspergillosis, including the target used in each case.

The most frequently diagnosed clinical forms using NGS, in descending order, were as follows: pulmonary aspergillosis (PA), invasive pulmonary aspergillosis (IPA), central nervous system (CNS) aspergillosis, patients with a lung transplantation, Bell’s palsy, malignant external otitis, a mediastinal *Aspergillus fumigatus* abscess, COVID-19-associated pulmonary aspergillosis (CAPA), acute lymphoblastic leukemia (ALL), infectious keratitis, bronchiectasis, endocarditis, COVID-19, and acute respiratory distress syndrome (ARDS). With the ISH method, the clinical forms reported were as follows: allergic fungal sinusitis (AFS), IPA, hematologic malignancies, acute invasive aspergillus rhinosinusitis (AIAR), invasive maxillary sinus aspergillosis (IMSA), invasive fungal infections (IFIs), and not determined (ND). For the NTS method, the most frequently diagnosed clinical forms were CNS aspergillosis and IPA. Likewise, for microarrays, the clinical forms reported were neutropenic patients, patients with severe asthma, and ND. For the RFLP method, the clinical form reported was neutropenic patients. For the LAMP method, the clinical forms reported were IA, chronic pulmonary aspergillosis (CPA), and ND (Appendix A).

On the other hand, the most frequently reported clinical forms associated with the use of PCR (including its different variants) as a diagnostic method for aspergillosis were as follows: For qPCR, they were invasive aspergillosis (IA), COVID-19-associated pulmonary aspergillosis (CAPA), rhinosinusitis, chronic pulmonary aspergillosis (CPA), and pulmonary aspergillosis (PA). For the multiplex PCR method, the most common were IA, CAPA, and aspergillosis. For the nested PCR method, the main forms reported were IA, CPA, and rhinosinusitis. For the endpoint PCR method, IA was the primary form (Appendix A).

The clinical samples used for NGS were as follows: BALF (21), blood (7), cerebrospinal fluid (CSF) (5), sputum (3), tissue (2), a transbronchial lung biopsy (TLBL) (1), pleural fluid (1), endobronchial ultrasound (1), and plasma (1). For NTS, the samples used were CSF and blood. For ISH, the samples were tissue (6), BALF (1), and sputum (1), while one study did not specify the sample used. For microarrays, the samples were blood (2), tissue (1), and BALF (1), and one study did not specify the sample used. For RFLP, the samples were blood (1). For LAMP, BALF (1), blood (1), and sputum (2) were used, and one study did not specify the sample used (Appendix A). For PCR, the different varieties of samples were as follows: BALF (35), serum (14), blood (8), tissue (6), sputum (4), plasma (4), endotracheal aspirate (4), CSF (3), pleural fluid (2), pericardial fluid (1), bronchial secretion (1), synovial fluid (1), wounds (1), and abscesses (1) (Appendix A).

In addition, the *Aspergillus* species identified with each of the molecular methods showed that *A. fumigatus*, *A. flavus*, *A. niger*, and *A. terreus* were the most frequently identified species with most of the reported molecular methods: NGS, NTS, ISH, microarrays, qPCR, multiplex PCR, RT-PCR, endpoint PCR, U-dHRM, and ddPCR. Meanwhile, other species such as *A. lentulus*, *A. oryzae*, *A. versicolor*, *A. terreus*, *A. sydowii*, *A. nidulans*, *A. glaucus*, and *A. citrinoterreus* were only identified by NGS or PCR (Appendix A).

## 4. Discussion

The literature reviewed in this work demonstrates that some molecular methods implemented as tools for diagnosing aspergillosis are highly sensitive and specific, capable of detecting and quantifying small amounts of deoxyribonucleic acid (DNA) and ribonucleic acid (RNA) directly from clinical samples.

It was also revealed that the most commonly employed methods were various PCR modalities and NGS. The use of PCR for detecting fungi has been known for many years, as its high sensitivity and specificity enable the detection of *Aspergillus* spp. DNA before the onset of clinical manifestations or obvious radiological findings. This is of great importance, particularly in immunocompromised patients. Furthermore, it has advantages over other diagnostic methods, such as cultures, as these other methods can take several days to provide results and may show a low sensitivity, especially if the patient is receiving antifungal treatment. Currently, several home PCR tests and even some commercially available assays are used for the molecular detection of *Aspergillus* spp. Among the most widely used commercial assays are AsperGenius^®^, MycAssay *Aspergillus*^®^, and MycoG ENIE^®^, which have the advantage of improved species-level identification and the simultaneous detection of antifungal resistance [146].

On the other hand, it is essential to consider that each of the PCR variants has its own particular advantages and disadvantages. The qPCR method involves no sample transfer, no new reagents are added, and no agarose gel is used to visualize the results. Thus, the chances of cross-contamination from the identical amplicons obtained in previous PCRs are significantly reduced. This technique allows for the identification of fungi at the genus or species level, based on probes or primers specifically designed to detect a known sequence of a particular fungus. qPCR has unique advantages because it allows for the real-time kinetic detection of amplified product accumulation through fluorescence intensity changes in a closed-tube environment, eliminating the need for post-amplification manipulation and, thus, significantly reducing the chances of cross-contamination [147]. qPCR methods represent an excellent alternative to existing standard culture methods, as they allow for the reliable detection and quantification of several pathogens [148]. Among the advantages of qPCR is its speed; no additional detection process is needed, since the amplification and detection process can be completed in 30–40 min. Another advantage of qPCR is that, by using closed systems, the risk of contamination decreases significantly; in addition, qPCR equipment has a very high capacity, since qualitative and quantitative tests, mutation determination, multiple PCR, etc., can be carried out in the same instrument. At the same time, conventional procedures require multiple teams [149].

An emerging alternative technique to qPCR that allows for the detection of *Aspergillus* in biological samples with high sensitivity and precision is ddPCR, which enables accurate quantification in samples with minimal fungal loads, such as those from the respiratory tract. Such loads could be missed by qPCR [1,2,4,5,6,9,58]. While standard qPCR is useful, it lacks sensitivity below its lower limit of detectable thresholds for individual microorganisms. For example, in cystic fibrosis and bronchiectasis, the airway ecology is a complex environment with multiple coexisting organisms. Therefore, the ability to detect organisms at ultra-low numbers would be very valuable, and ddPCR may offer this attractive alternative to qPCR in certain clinical settings [53,54,55,56,57,58,150]. This is the case for allergic bronchopulmonary aspergillosis (ABPA) in cystic fibrosis, where ultra-low numbers of fungi are likely present in the airways, but are currently not detected by standard PCR methods. Furthermore, a key advantage of ddPCR over traditional TaqMan qPCR is the direct quantification of a target microbe without the need for a standard curve or controls. This improves the reproducibility and accuracy by eliminating the reliance on quantitative reference materials, whose quantification, origin, batch, storage, and handling conditions can influence the qPCR results for biological samples [59]. Furthermore, the attributes offered by ddPCR can be beneficial in the current era, where microbiomes, including the mycobiome, are gaining greater importance and relevance for understanding the pathogenesis, disease progression, and consequences of various respiratory diseases [1,4,46,49,60,62,66,67]. Beyond fungi, the usefulness of ddPCR can likely be extended to a wide range of microorganisms, organ systems, and human diseases if appropriately applied to the right sample, in the right setting, and to address a specific clinical question [151]. However, despite the multiple advantages that this technique offers, only one reference was found related to this technique in this review. The authors of this reference discuss the good performance and usefulness of this technique when applied to diagnose aspergillosis, particularly IPA, from BALF. Perhaps because it is a relatively new technique, it is a matter of time before its application becomes wider in the diagnosis of aspergillosis.

Although the U-dHRM method is promising for its throughput and speed, as well as its ability to simultaneously identify and quantify clinically relevant mold pathogens in polymicrobial samples and to detect emerging opportunistic pathogens, it is still early days for an evaluation, as a reliable cut-off value needs to be established to improve the specificity of infection versus colonization and, therefore, the accuracy of this method [26]. However, it has a limited application for the diagnosis of aspergillosis, as only one reference related to the detection of mold-related lung infections from BALF was identified in this review.

Another variant is the multiplex PCR, which has the advantages of combining the sensitivity and speed of the method while eliminating the need to evaluate clinical samples independently for each fungus. It allows for the optimization of reagent use and reduces the diagnostic time; however, it characteristically requires careful optimization. Another advantage of this method is the detection of double or multiple coinfections; these results could have important implications for epidemiology and treatment [23].

On the other hand, nested PCR has the disadvantage of requiring greater manipulation, which raises the risk of contamination with foreign DNA.

The results of this review showed that qPCR was the most widely used variant for the diagnosis of aspergillosis [92,93,94,95,96,97,98,99,100,101,102,103,104,105,106,107,108,109,110,111,112,113,114,115,116,117,118,119], followed by multiplex PCR [120,121,122,123,124,125,126,127,128,129,130,131], nested PCR [132,133,134,135,136,137,138,139,140,141,142], RT-PCR [143], PCR [144,145], U-dHRM [26], and ddPCR [27].

Despite the advantages of various PCR variants, the literature shows conflicting results regarding the best biological sample to use for each. In this review, it was clear that most references found bronchoalveolar lavage (BAL) the most useful [94,100,101,104,109,111,112,113,114,115,116,117,118,121,123,124,127,128,129,130,132,133,134,136,137,139,140,141,142,145], while others mentioned that serum, plasma, or tissue samples can also be used. However, it is very important to consider the patient’s clinical context, as the choice of sample depends on it. Additionally, it was found that the most common samples for PCR in different modalities for AI cases were BALF [94,100,101,104,109,111,112,113,114,115,116,117,118,121,123,124,127,128,129,130,131,132,133,134,135,136,137,138,139,140,141,142,145], serum [92,93,95,96,99,100,106,109,120,125,137], and blood [98,99,103,134,139,143]. Similarly, for other clinical forms such as CAPA [105,119,126] and CPA [107,135], the clinical sample used was BALF, whereas for rhinosinusitis [103,139], tissue samples were used.

To validate PCR methods in the diagnosis of aspergillosis, a group of experts founded the Fungal PCR Initiative (FPCRI) (www.fpcri.eu (accessed on 6 May 2025)) to standardize the nucleic acid extraction protocol from whole blood, plasma, and serum, noting that it was the rate-limiting step in PCR for *Aspergillus*. In addition, the Quality Control for Molecular Diagnostics (QCMD) and the Fungal Diagnostic Laboratory Consortium (FDLC, based at Johns Hopkins, USA) have joined these groups to achieve this goal [149].

The results of this review also show that NGS is a handy tool for the diagnosis of aspergillosis, and one of its main advantages is that the sequences obtained by NGS provide a large amount of data, allowing for the identification of different microorganisms that can be detected simultaneously. The reliability of the sequence data exceeds that of traditional morphological and physiological assays for microbial identification [152]. Likewise, NGS has a significant advantage for diagnosing rare or mixed infections, especially in samples with negative cultures [153,154].

NGS allows for the detection of coinfections of *Aspergillus* with other microorganisms, such as cholesteatoma [91], *Legionella pneumoniae* [155], Stevens–Johnson syndrome (SFTS) associated with IPA [58], and mediastinal abscesses [57]. It has also been helpful in the identification of *Aspergillus* associated with central nervous system diseases [62,156,157], highlighting the usefulness of CSF for the non-invasive identification of cerebral aspergillosis (CA), since *Aspergillus* is rarely detected in CSF cultures after a suspected intracranial fungal infection [157,158].

NGS is highly relevant in identifying mixed infections, which primarily occur in immunosuppressed patients [159]. It can also be an important tool for preventing misdiagnoses and underdiagnoses [160].

In one case, Yan et al. [47] detected *M. tuberculosis* and *A. fumigatus* using conventional methods; however, using NGS, they also detected *N. nova* and cytomegalovirus, in addition to *M. tuberculosis* and *A. fumigatus*.

Likewise, in a patient with pulmonary infection and diabetes mellitus, galactomannan and 1,3-β-D-glucan tests were performed and came back positive; however, acid-fast staining and a CSF culture were negative, and the detection of *M. tuberculosis* DNA in the CSF was positive. To confirm the results, NGS was performed, which confirmed a pulmonary coinfection with *M. tuberculosis* and *A. lentulus* [64].

The sensitivity of NGS for diagnosing mixed lung infections is seven times higher than that of conventional tests (97.2% vs. 13.9%). Similarly, another study found that the rate of confirmed mixed lung infections detected by NGS was four times greater than that identified by conventional methods [161]. Another significant benefit of NGS over other microbiological tests is its ability to identify a wide range of pathogens, which permits the detection of mixed infections with a single sample from patients with pulmonary aspergillosis. Moreover, it has been reported that 50% of patients with pulmonary aspergillosis did not receive anti-*Aspergillus* medications until after the NGS results were reported, demonstrating that NGS is valuable for diagnosing and guiding the treatment of pulmonary aspergillosis.

The diagnostic performance of NGS in non-neutropenic patients with pulmonary aspergillosis has also been studied, revealing that the sensitivity of NGS is significantly higher than that of conventional etiological methods and serum (1,3)-β-D-glucan, but with a relatively low specificity. Furthermore, the diagnostic efficacy of NGS for confirmed IPA has been evaluated, and it has been found that NGS for bronchoalveolar lavage fluid (BALF) has a high diagnostic efficacy for confirmed IPA, superior to *Aspergillus* culture in sputum or BALF and GM testing in blood or BALF [56].

In a retrospective study by Yang et al. [162], the usefulness of cultures and NGS in diagnosing rare pathogens was compared in children with hematological diseases. They found that the positivity rate of NGS for bacteria and fungi was significantly higher than that of the cultures (57.2% vs. 12.5%, *p* < 0.01), and the concordance (i.e., the detection of at least one of the same pathogens) between NGS and the cultures was 90.9% (10/11). Their results indicated that NGS not only shows a good consistency with traditional methods, but is also an effective diagnostic tool with a high positivity for rare pathogens.

It is essential to note that NGS can also detect pathogens even when an empirical anti-infective treatment has been previously administered, suggesting that NGS is rarely affected by such treatments [163].

Another example illustrating the utility of NGS involved an infant with chronic granulomatous disease (CGD) who had IPA. A lung tumor was initially suspected, and blood, sputum, and BAL cultures were performed; however, the results were negative and did not identify the pathogen. Therefore, NGS with BALF was used, and *A. fumigatus* was identified as the causative agent [164].

The effectiveness of NGS has been evaluated using various clinical samples as a diagnostic tool for different diseases. Shi et al. [49] used BAL to identify pathogens in rheumatic patients with suspected pneumonia, comparing the NGS results with combined microbiological tests, including the following: a culture for bacteria and fungi; special staining for *Mycobacterium*, *Cryptococcus*, and P. *jirovecii*; serological antibody tests for atypical respiratory pathogens; antigen detection for fungi, influenza A/B, and *Legionella pneumophila*; a direct examination for fungi; and PCR testing for atypical respiratory pathogens. Their findings demonstrated that NGS was superior to combined microbiological testing (CMT), particularly in viral detection, which aligns with previous reports [165,166,167]. It is important to note that the low accuracy of CMT may be related to the fact that physicians do not request PCR testing for all viruses in every patient, reflecting a real clinical situation: because of the wide variety of viruses, it is difficult to analyze each one individually, and hospitals often lack routine identification methods for all viruses, including rare ones.

Other studies have compared the performance of NGS with cultures and have shown that NGS has a higher sensitivity (50.7%) and specificity (85.7%); however, they showed a statistically different sensitivity according to the biological sample used (blood, BAL, or sputum) [163,164]. Likewise, other studies have shown that BAL is the most suitable sample, with a higher fungal load than that of sputum or blood, exhibiting a sensitivity and specificity of 84.4% and 85.3%, respectively, which is helpful in diagnosing pulmonary fungal infections [47,168].

Although identifying *Aspergillus* in BALF through NGS is a good option for diagnosing IPA, in clinical practice, patients with severe infections may experience complications such as severe hypoxemia, bleeding, or thrombocytopenia, making invasive procedures such as a bronchoscopy difficult to perform. An alternative is to use peripheral blood, as it can be used to quickly detect bacterial and fungal infections simultaneously and help tailor appropriate treatment plans [169]. However, its effectiveness for detection still needs to be confirmed by large-scale clinical trials.

Significant challenges remain in using NGS for routine patient care, especially concerning its reduced sensitivity in detecting pathogens in clinical samples due to a high background of nucleic acids from the human host (e.g., in tissue biopsies) [170].

Some studies have shown that NGS has a higher detection rate than cultures [167,171], while others have observed a comparable detection rate between NGS and conventional tests [172]. Particularly in the detection of *Aspergillus* spp., NGS has shown an increased sensitivity by approximately 15% compared with cultures, but this sensitivity was still lower than the combination of the GM test and a culture. Therefore, in current clinical practice, NGS needs to be combined with a fungal antigen test due to its low sensitivity for *Aspergillus* spp.

In most NGS works, the most commonly used platform was Illumina, followed by Ion Torrent; however, it is important to mention that some references do not mention the type of platform used, which does not enable a comparison or analysis of the differences between them. To standardize this method, it would be interesting to evaluate the most appropriate type of platform in order to obtain a greater precision and sensitivity in sequencing and data analyses.

The TGS method has also been used for the clinical diagnosis of aspergillosis. However, despite the advantages offered by third-generation sequencing, it has its disadvantages. Perhaps among the most relevant are that it requires more computing equipment for processing, the required specialized tools are less available, and it has a higher cost per sample, making it inaccessible to many clinical diagnostic laboratories. We only found two articles that used this technology to diagnose *Aspergillus* from clinical samples [74,75], and it is probably a matter of time before its use becomes widespread. Likewise, sequencing methods have used the 18S rRNA/ITS regions. These genes constitute a conserved component of eukaryotic genomes. These include both conservative and variable regions (V1 to V9, excluding V6, which is relatively conservative), reflecting interspecies differences among eukaryotes such as fungi. 18S rRNA sequencing remains the gold standard for eukaryotic molecular phylogeny. The sequencing of the 18S rRNA gene is routinely employed for the identification, classification, and quantification of microbes in complex biological mixtures. On the other hand, internal transcribed spacer (ITS) sequences, which diverge in their target sequences, are located in the transcriptional interstitial regions of ribosomal RNA (rRNA) gene constituents, typically spanning the ITS1 and ITS2 subregions. Their sequences are usually shorter and exhibit a high variability, whereas 18S rRNA sequences are notably longer and relatively conserved [173,174]. There are differences in species identification with these markers; when using short-read-length 18S rRNA sequencing with NG-Illumina or full-length 18S rRNA sequencing with PacBio sequencing or Nanopore sequencing, there are differences between second- and third-generation sequencing. The latter seems to be superior to its counterparts, as the entire 18S region is covered. In this way, species can be more accurately identified, as each variable subregion has different capacities to represent species, so an accurate taxonomic classification can only be achieved by sequencing all the variable regions of the 18S gene [175]. Another disadvantage is that, in some cases, closely related species have been observed to share the same ITS sequence [176,177,178]. In such cases, intersterility barriers have evolved more rapidly than the corresponding ITS divergence [179]. This is known to be the case in many genera, such as *Aspergillus* and *Penicillium* [180], where additional DNA markers are needed to separate between closely related taxa.

Further, the bioinformatics approach to second-generation metagenomic sequencing offers several advantages, including the ability to produce short, but highly accurate, reads with high fidelity, allowing for the identification of specific variants. There are several programs that optimize short reads, such as QIIME2, MetaPhlAn, and Kraken2. The costs of this approach are also more affordable. Its disadvantages include the difficulty in assembling complete genomes and detecting structural variants or long genes. Importantly, it relies on reference databases, which limit the identification of new organisms. On the other hand, third-generation metagenomics has advantages such as the generation of long reads, the ability to assemble complete genomes, and the detection of structural variants and mobile elements. Epigenetic analyses are also possible thanks to the direct reading of modifications. This approach also has a lower amplification bias because the platforms do not require PCR. The disadvantages include a higher error rate per reading, although this can be corrected through hybrid algorithms (which allow for the integration of short high-fidelity reads, such as the Illumina platform, with long low-fidelity reads, such as the Nanopore/PacBio platforms) to improve the accuracy of the assembly and taxonomic identification. This method also represents a greater demand regarding computing equipment for processing and error correction; additionally, there is a lower availability of specialized tools, and it has a higher cost per sample [16,17,18].

The ISH, LAMP, microarray, and PCR-RFLP methods were the least commonly used, possibly due to their specific disadvantages. A weakness of the ISH technique is its high degree of background staining and the low intensity or focal staining present in some cases. A weakness of the ISH technique is its high degree of background staining and the low intensity or focal staining present in some cases. These limitations are likely due to several factors: fixatives are detrimental to the in situ hybridization procedure [181]; additionally, the thickness of the mycelium can influence the result in situations with a lower signal intensity (i.e., a larger area of signal distribution within the section per organism); the presence of extensive necrosis in some cases can also have a detrimental effect on detection; furthermore, necrotic tissue can interfere with nucleic acid hybridization; and finally, extensive tissue degradation can result in the release of endogenous nucleases [182]. Another significant disadvantage of LAMP is its high risk of carryover contamination, which often leads to false-positive results in negative controls [183,184]. Another limitation is that the turbidimetric and colorimetric determination of the LAMP reaction is subjective when determined solely by visual observation, as it depends on color perception [185]. Furthermore, when applied to a gel, the LAMP product shows a stair-like or blobby pattern rather than a single band as in PCR, so the identification of the band size on a gel is not possible with LAMP [186]. A disadvantage of the microarrays is their low sensitivity, which currently makes it difficult to use them for the direct detection of microorganisms in clinical samples. Therefore, due to their low diagnostic yield, microarrays should not be used as a routine diagnostic method or as a screening tool [85,187]. With respect to PCR-RFLP, the reproducibility is considered a disadvantage of this technique due to the variable intensity of the bands obtained, which can be attributed to small variations in the various procedural steps that affect the final peak intensity [13,188,189]. For the diagnosis of aspergillosis, PCR-RFLP has proven useful for detecting *Aspergillus* at the genus level in blood samples from neutropenic patients [87].

## 5. Conclusions

This review showed that the most commonly used molecular methods for the clinical diagnosis of aspergillosis were PCR in different variants and NGS; however, despite the advantages that PCR variants have shown, their standardization has not yet been achieved due to the lack of solid scientific evidence due to the variability in study designs, the primers and probes used, the DNA extraction and amplification protocols, and the scarcity of commercial assays.

Regarding NGS, there is also no consensus on whether its advantages for the detection of fungi are sufficient to use it as the sole tool for the diagnosis of aspergillosis.

The most widely used methods in the diagnosis of aspergillosis are qPCR and NGS; however, each method has advantages and disadvantages. qPCR has demonstrated the highest sensitivity and specificity for clinical samples (such as blood, serum, bronchoalveolar lavage [BAL], tissue, or sputum), since it detects specific sequences. Furthermore, the validation of this method shows greater progress towards achieving this goal. On the other hand, NGS is not a targeted technique, since it sequences all the genetic material present. Furthermore, the sensitivity for the detection of pathogens decreases when clinical samples are used due to a high background of nucleic acids present in the human host (e.g., in tissue biopsies). Likewise, an advantage of qPCR over NGS is the time to obtain results, since obtaining results is faster with qPCR than with NGS.

## Figures and Tables

**Figure 1 jof-11-00720-f001:**
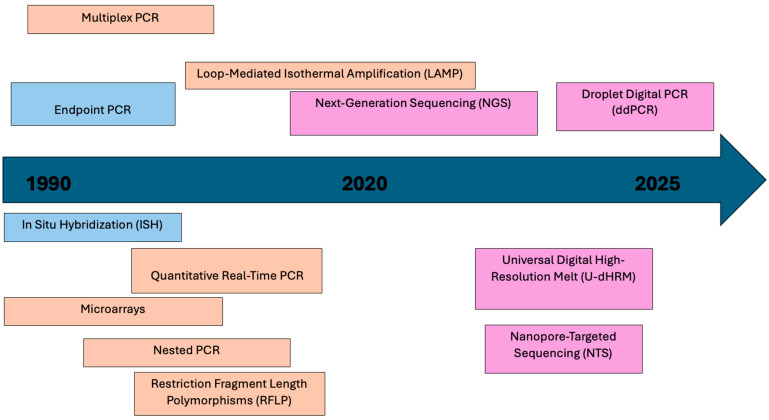
Timeline illustrating the progress of molecular methods in the diagnosis of aspergillosis from 1990 to 2025.

## Data Availability

The data are contained within the article and Appendix A.

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
