# Peer review of "HTS and PCR Methods Are the Most Used in the Diagnosis of Aspergillosis: Advantages over Other Molecular Methods"

_jof, 2025, doi:10.3390/jof11100720_

Round 1
Reviewer 1 Report (Previous Reviewer 1)
The manuscript clearly and thoroughly describes the various molecular tools used to identify aspergillosis. The review is well-written and easy to read.
line 404. Change A. fumigatus to A. fumigatus
Author Response
REVISOR 1
Major comments
The manuscript clearly and thoroughly describes the various molecular tools used to identify aspergillosis. The review is well-written and easy to read.
Detailed comments
line 404. Change A. fumigatus to A. fumigatus
Answer: It was corrected in the text.
Reviewer 2 Report (Previous Reviewer 2)
Overall, the manuscript is highly relevant for publication. However, although the suggested changes were implemented, they made the text tiring to read and lengthy. Nevertheless, the scientific content is rich in detail and contributes to the understanding of aspergillosis diagnosis.
The abstract is redundant. Sentences such as "this review shows..." are repeated throughout the text. I suggest revising the abstract. The keywords are not sufficient; try to select ones that will have an impact on searches for manuscripts detailing diagnostic techniques for aspergillosis. Please revise the keywords.
In the introduction, regarding the descriptions of molecular methodologies, the text feels lengthy and tiring for the reader. There is relevant information, which I appreciated, but the section is overly long. I suggest revising it.
Figure 1 is very descriptive, an excellent illustration. I believe the suggested changes were accepted and have added relevant information to the manuscript.
Author Response
REVISOR 2
Major comments
Overall, the manuscript is highly relevant for publication. However, although the suggested changes were implemented, they made the text tiring to read and lengthy. Nevertheless, the scientific content is rich in detail and contributes to the understanding of aspergillosis diagnosis.
Detailed comments
The abstract is redundant. Sentences such as "this review shows..." are repeated throughout the text. I suggest revising the abstract. The keywords are not sufficient; try to select ones that will have an impact on searches for manuscripts detailing diagnostic techniques for aspergillosis. Please revise the keywords.
Answer: We appreciate your comments and have decided to submit the new English version of the manuscript for review (Edited by MDPI). Additionally, following your suggestion, we revised the keywords and added new ones.
In the introduction, regarding the descriptions of molecular methodologies, the text feels lengthy and tiring for the reader. There is relevant information, which I appreciated, but the section is overly long. I suggest revising it.
Answer: We appreciate your feedback and have removed some parts of the introduction. However, although this section is still long, we hope that this new version will be more understandable for readers.
Figure 1 is very descriptive, an excellent illustration. I believe the suggested changes were accepted and have added relevant information to the manuscript.
Reviewer 3 Report (New Reviewer)
The manuscript by Castro-Fuentes et al. provides a review of an important topic in medicine, which is the molecular diagnosis of aspergillosis, and the advantages of metagenomic sequencing and PCR over other methods.
Although the review covers a lot of aspects of the molecular diagnosis of aspergillosis, the review is not complete in terms of discussing the methods of Aspergillus species and bioinformatic approaches to sequencing data.
The manuscript needs to be significantly improved before the decision on publication.
1. The term “mNGS” may not be the most appropriate. The abbreviation NGS generally refers to second-generation sequencing technologies, whereas third-generation platforms, such as Nanopore and PacBio sequencing, are also widely used. I recommend changing the title of the review and adding a discussion of the differences between second and third-generation sequencing technologies for the metagenomic detection of Aspergillus species. The review omits a discussion of the distinct differences in effectiveness between these approaches, which would be valuable for readers.
"mNGS" can be replaced with "metagenomic sequencing".
2. There is a very important approach for the molecular detection of Aspergillus species not discussed by the authors, which is the mycobiota metaprofiling based on the ITS/18S rRNA amplicon high-throughput sequencing. The lack of discussion of this method in the review weakens its comprehensiveness, as this approach is widely applied in both clinical and environmental studies to characterize fungal communities and can provide valuable insights into Aspergillus prevalence and diversity.
It is important to discuss that second-generation sequencing technologies cannot provide species-level detection of representatives of the Aspergillus genus due to the limitations in the length of amplicon fragments.
An example of the paper presenting the results of species-level ITS/18S metaprofiling with third-generation sequencing technologies: https://doi.org/10.1093/biomethods/bpaa026
3. Also, the review does not contain a discussion of the advantages and disadvantages of different bioinformatic approaches for the metagenomic data analysis, which can affect the results of diagnostics. Addressing this aspect would be highly beneficial, as differences in pipelines, reference databases, and analytical strategies can substantially influence taxonomic resolution, detection sensitivity, and overall interpretability of the findings.
For example, taxonomic identification of raw sequencing data and assembled metagenomes can result in different outcomes, as read-based approaches may offer higher sensitivity, while assembly-based methods can improve specificity and facilitate functional characterization. These methodological differences can directly influence diagnostic accuracy and should be addressed in the review.
4. Why is most of the text highlighted in yellow?
5. Please consider writing taxa names in italics, at least on the genus and species levels.
Author Response
REVISOR 3
Major comments
The manuscript by Castro-Fuentes et al. provides a review of an important topic in medicine, which is the molecular diagnosis of aspergillosis, and the advantages of metagenomic sequencing and PCR over other methods.
Although the review covers a lot of aspects of the molecular diagnosis of aspergillosis, the review is not complete in terms of discussing the methods of Aspergillus species and bioinformatic approaches to sequencing data.
The manuscript needs to be significantly improved before the decision on publication.
Detailed comments
- The term “mNGS” may not be the most appropriate. The abbreviation NGS generally refers to second-generation sequencing technologies, whereas third-generation platforms, such as Nanopore and PacBio sequencing, are also widely used. I recommend changing the title of the review and adding a discussion of the differences between second and third-generation sequencing technologies for the metagenomic detection of Aspergillusspecies. The review omits a discussion of the distinct differences in effectiveness between these approaches, which would be valuable for readers.
"mNGS" can be replaced with "metagenomic sequencing".
Answer: We appreciate your valuable comments and have included analysis of third-generation technology.
- There is a very important approach for the molecular detection of Aspergillusspecies not discussed by the authors, which is the mycobiota metaprofiling based on the ITS/18S rRNA amplicon high-throughput sequencing. The lack of discussion of this method in the review weakens its comprehensiveness, as this approach is widely applied in both clinical and environmental studies to characterize fungal communities and can provide valuable insights into Aspergillusprevalence and diversity.
It is important to discuss that second-generation sequencing technologies cannot provide species-level detection of representatives of the Aspergillus genus due to the limitations in the length of amplicon fragments.
An example of the paper presenting the results of species-level ITS/18S metaprofiling with third-generation sequencing technologies: https://doi.org/10.1093/biomethods/bpaa026
Answer: We reiterate our gratitude for your valuable comments and, in response to these, we have included them in the discussion section.
- Also, the review does not contain a discussion of the advantages and disadvantages of different bioinformatic approaches for the metagenomic data analysis, which can affect the results of diagnostics. Addressing this aspect would be highly beneficial, as differences in pipelines, reference databases, and analytical strategies can substantially influence taxonomic resolution, detection sensitivity, and overall interpretability of the findings.
For example, taxonomic identification of raw sequencing data and assembled metagenomes can result in different outcomes, as read-based approaches may offer higher sensitivity, while assembly-based methods can improve specificity and facilitate functional characterization. These methodological differences can directly influence diagnostic accuracy and should be addressed in the review.
Answer: We appreciate your valuable comments and suggestions for enriching and improving this work, and we have included in the discussion the advantages and disadvantages of different bioinformatics approaches for the analysis of metagenomic data.
- Why is most of the text highlighted in yellow?
Answer: Due to the reviewers' suggestions in the previous version, the content of the text was modified in its structure, which was the reason for marking it in yellow to highlight the changes.
- Please consider writing taxa names in italics, at least on the genus and species levels.
Answer: We apologize for these errors, the manuscript was reviewed and corrected.
Round 2
Reviewer 3 Report (New Reviewer)
The authors addressed all the comments in a good manner.
I still recommend correcting the title of the review. The use of the term “NGS” is misleading, as the review also covers third-generation sequencing technologies. Since “NGS” is commonly associated with second-generation platforms (e.g., Illumina), it does not adequately reflect the broader scope of the article.
I suggest that the authors instead use the term “HTS” (high-throughput sequencing). This terminology is more inclusive, as it encompasses both second- and third-generation sequencing technologies, and is widely accepted in the literature when referring to sequencing approaches that generate large volumes of data. Using “HTS” would therefore make the title more accurate, precise, and consistent with the content of the review.
Author Response
Reviewer 3
I still recommend correcting the title of the review. The use of the term “NGS” is misleading, as the review also covers third-generation sequencing technologies. Since “NGS” is commonly associated with second-generation platforms (e.g., Illumina), it does not adequately reflect the broader scope of the article.
I suggest that the authors instead use the term “HTS” (high-throughput sequencing). This terminology is more inclusive, as it encompasses both second- and third-generation sequencing technologies, and is widely accepted in the literature when referring to sequencing approaches that generate large volumes of data. Using “HTS” would therefore make the title more accurate, precise, and consistent with the content of the review.
Answer: We appreciate your comment and contributions, as they were very valuable for improving the work. After carefully reviewing the manuscript, we agree to change the title of the work. In the new version of the manuscript, the changes are highlighted in gray.
This manuscript is a resubmission of an earlier submission. The following is a list of the peer review reports and author responses from that submission.
Round 1
Reviewer 1 Report
1. Page 2, lines 43-45. Reference 2 is from 2020. It is advisable to include a new reference published in 2024. The old reference can be replaced, and the new classification described in 2024 can be included in the new version. (doi: 10.3114/sim.2024.107.01) "A review of recently introduced Aspergillus, Penicillium, Talaromyces and other Eurotiales species"
2. The discussion is very long and contains information not necessarily related to the aim of the manuscript, so it would be important to simplify the discussion and focus on addressing the title of the work (mNGS and PCR methods are the most useful in the diagnosis of 2 aspergillosis: advantages over other molecular methods). This applies to all discussion topics.
An example of the above is the discussion of PCR-RFLP (Page 16, lines 262-282). It has the two initial paragraphs describing the technique, but only two lines that can be related to the title of the manuscript.
3. The conclusions are quite simple; they could be complemented by an attempt to analyze (based on what has been seen in the referenced articles) the perspective of these tools for the diagnosis of Aspergillus infection. Additionally, and considering the manuscript title: What are the advantages of mNGS and PCR-based methods over other molecular methods? This could be clearer in the conclusion
Tables 1 and 2 are very long, but although they are necessary, they may be difficult to read and include in the final version of the manuscript. Consider splitting the tables
Reviewer 2 Report
Dear authors,
This review provides relevant information about the biological molecular techniques; however, the manuscript needs to be revised for a possible publication in the Jof.
Overall, the manuscript is well written, but it lacks depth of information. The tables are too extensive, and in my opinion, this may discourage the reader.
This review does not indicate which methodology is most suitable to support the diagnosis of Aspergillosis. It also does not assess which NGS sequencing platform was the most sensitive and specific for diagnosing the disease.
The conclusion of the review fails to present any definitive statements. It does not specify which technique is appropriate, nor does it summarize the authors’ observations when comparing the different molecular methods. The conclusion should clearly describe which techniques are more suitable and summarize the key findings from the comparative analysis of the molecular approaches.
Best regards.
Dear authors,
This review presents relevant content; however, substantial revisions are needed for it to be considered for publication in the Journal of Fungi (JoF). Below are specific comments and suggestions:
1. Abstract
-
The abstract lacks essential information on which diagnostic methods—mNGS or various PCR modalities—are most appropriate for the diagnosis of Pulmonary Aspergillosis.
-
Please revise to include clearer conclusions based on your findings.
2. Keywords
-
Include terms such as mNGS or metagenomic, PCR, RFLP, MLST, and real-time PCR to enhance visibility in databases and search engines.
3. Tables
-
Tables 1 and 2 are overly long. I recommend shortening them or moving them to supplementary material. Though informative, their length may overwhelm readers and reduce engagement.
4. Methodological Details (mNGS & PCR)
-
Clearly state which sequencing platforms were used in each study (e.g., Illumina, Nanopore, Pyrosequencing).
-
Specify how mNGS was performed. Was it a broad, untargeted approach or directed specifically at Aspergillus spp.?
-
mNGS is generally considered more sensitive than PCR-based techniques. How was this demonstrated or quantified in the reviewed studies?
-
Were there challenges in analyzing the sequencing data? If so, describe them in the discussion.
5. Sample Types
-
Indicate which clinical samples (e.g., BAL, sputum) are most appropriate for each technique.
-
Discuss the potential risks of contamination with each sample type.
6. Discussion
-
The discussion should include the challenges reported in the use of mNGS (e.g., high cost, complexity, contamination, time to results).
-
Also, highlight gaps or inconsistencies found across the studies.
7. Critical Gaps
-
The review does not clearly indicate which methodology is best suited to support Aspergillosis diagnosis.
-
It also does not evaluate which NGS platform proved to be the most sensitive or specific.
-
These are crucial points and must be directly addressed in the manuscript.
8. Conclusion
-
The conclusion is currently lacking and fails to present a synthesis of findings.
-
What technique is recommended? Which was more sensitive or specific?
-
What did the authors observe in their comparison of molecular methods?
-
The conclusion should directly answer these questions and provide a strong summary of the key takeaways.
9. Overall Impression
-
The manuscript is well written in terms of language and structure, but it lacks depth and critical analysis.
-
The extensive tables and limited discussion may reduce reader motivation and impact.
-
With improvements in content, focus, and interpretation, this review has potential to make a valuable contribution to the field.
- Create a flowchart that details when this technique was first used for the diagnosis of Aspergillosis—a timeline
Reviewer 3 Report
The manuscript titled "mNGS and PCR methods are the most useful in the diagnosis of aspergillosis: advantages over other molecular methods" addresses an important and timely topic, the use of molecular and metagenomic next-generation sequencing (mNGS) methods in the diagnosis of aspergillosis. While the subject matter is highly relevant, I have several concerns regarding the clarity, comprehensiveness, and critical analysis within the review.
- The stated aim of the manuscript is to review the molecular methods used in the diagnosis of aspergillosis and to highlight the advantages of molecular and mNGS techniques. However, the scope of the review remains somewhat ambiguous throughout the text. It would benefit from a clearer definition of its focus and objectives early in the manuscript—whether it intends to be a broad overview, a comparative evaluation, or a clinically oriented guide to test selection.
- The review omits key emerging techniques such as Droplet Digital PCR (ddPCR) and Universal Digital High-Resolution Melt (U-dHRM), which are increasingly relevant in fungal diagnostics, while including outdated methods like PCR–Restriction Fragment Length Polymorphism (PCR-RFLP), End-point PCR, or declining ones (microarrays), without giving sufficient context regarding their current relevance and use.
- One of the main limitations of the manuscript is the lack of alignment between diagnostic tools and clinical contexts. Rather than listing the most frequently diagnosed clinical forms by each method, it would be more informative to evaluate which techniques are most effective in specific clinical settings (e.g., neutropenic patients, ICU patients, patients receiving mold-active antifungal therapy) and with different types of specimens (e.g., plasma, BALF). This would enhance the practical value of the review for clinicians and laboratory specialists.
- The manuscript would greatly benefit from a more critical assessment of the advantages, limitations, and appropriate applications of each diagnostic method. While it summarizes numerous studies, it does not provide sufficient interpretation or synthesis of the findings. Furthermore, there are recent high-quality reviews on the same subject that offer a more nuanced perspective, and the manuscript would benefit from positioning itself relative to this existing literature.
- While the overall language is understandable, the manuscript would benefit from careful editing for style to enhance clarity and fluency. However, this is a secondary concern compared to the issues raised above.
Tables.
- Presenting the reviewed studies sequentially in the tables limits the manuscript’s accessibility and comparative value. A tabular summary comparing the mNGS with the various molecular and PCR-based methods—highlighting their performance characteristics, sample requirements, turnaround times, and clinical applicability—would provide a clearer and more reader-friendly format.